# Joint Attention and Its Relationship with Autism Risk Markers at 18 Months of Age

**DOI:** 10.3390/children9040556

**Published:** 2022-04-13

**Authors:** Maite Montagut-Asunción, Sarah Crespo-Martín, Gemma Pastor-Cerezuela, Ana D’Ocon-Giménez

**Affiliations:** 1Department of Neuropsychobiology, Methodology, and Basic and Social Psychology, Universidad Católica de Valencia San Vicente Mártir, 46100 Burjassot, Valencia, Spain; maite.montagut@ucv.es; 2Department of Basic Psychology, Universitat de València, 46010 Valencia, Spain; sacres3@alumni.uv.es (S.C.-M.); ana.ocon@uv.es (A.D.-G.)

**Keywords:** joint attention, ASD, early symptomatology, ASD risk

## Abstract

(1) Joint attention is the ability to coordinate attention to share a point of reference with another person. It has an early onset and is a clear indicator of understanding the representations of others, and it is essential in the development of symbolic thought and the acquisition of language. Deficiencies in this prelinguistic early communication skill are strong markers of the risk of autism spectrum disorder (ASD); (2) this longitudinal study aimed to evaluate joint attention skills in a group of 32 infants at two developmental moments (8 and 12 months) in order to explore whether their performance on this skill was related to the presence of early signs of ASD at 18 months. Logistic multiple regressions were carried out for the data analysis; (3) results of the analysis showed that the variables of initiating joint attention at 8 months and responding to joint attention at 12 months were linked to the risk of ASD at 18 months of age; (4) in conclusion, early joint attention skills had a pivotal role in defining early manifestations of ASD.

## 1. Introduction

### 1.1. Joint Attention

Joint attention is the ability to coordinate attention to share a point of reference with another person [1,2,3]. Very young children show this skill from the first months of life, given that it has an early onset at around 8–12 months [4]. It is a clear indicator of understanding the representations of others, and it plays a primary role in the emergence of symbolic thought and the development of language [2,4,5,6]. Joint attention is a triadic ability that involves three elements: the adult, the baby, and the object. A typical situation where joint attention can be observed would be in an interaction between an adult and a baby where one of them is holding a toy. Both participants in the episode are attending to the same object and are aware that they have a common focus of attention.

We can differentiate between the *declarative function* and the *imperative function* of joint attention [2,7], which correspond to different facets of social-cognitive development and should be viewed differently. The declarative function of joint attention refers to sharing a common focus in order to communicate interest in the object. The imperative function of joint attention is used to accomplish an objective; that is, to obtain an object or make someone perform an action that is useful in achieving a goal [2]. Often, the term *joint attention* is restricted to the declarative use of this communicative behavior, whereas the term *behavioral request* refers to the imperative use of joint attention [8]. We used the term *initiating joint attention* when the episode was initiated by the child (see Figure 1b), and *responding to joint attention* if the child responded to an invitation from the adult (see Figure 1a). The same terms could be applied to initiating and responding to a behavioral request.

### 1.2. Joint Attention and ASD Diagnosis

Deficits in this early prelinguistic communication skill, especially difficulties in declarative purpose joint attention, are clear early markers of risk of autism spectrum disorder (ASD) [5,9,10]. Although there is a strong consensus that joint attention disturbances are a solid early marker of the risk of developing the disorder, there is disagreement among different authors about when these signs first start to emerge.

There have been inconsistent findings regarding whether joint attention disturbances can be observed before the age of six months, given that the early signs can indicate risk of ASD or risk of other developmental disorders, such as language disorders. Therefore, the results must be interpreted with caution [9,11,12,13,14,15,16]. Firm behavioral markers begin to be recognizable only after the first birthday [5,9,10]. According to the literature, many infants who show signs at the age of 14 months can be diagnosed reliably [17,18], and at around 18 months old, solid markers of autism can be detected in most cases [19,20,21,22].

In addition, differences in early joint attention are essential for later social-cognitive development and learning [9,23], particularly in children with autism. Because their motivation to participate in social interactions is generally lower, they are less likely to display joint attention behaviors, which probably attenuates their later social development and especially their language acquisition [1,24,25,26,27,28,29,30]. Therefore, it is important to better understand how this early communication skill develops both in children with neurotypical development and in children with autistic development. Being able to promptly detect the first alterations would allow professionals to address them in early stages, improving the prognosis of these children and positively impacting their quality of life and that of their families. Early interventions that target joint attention seem to produce positive results in improving the social performance of these children.

### 1.3. ASD and Early Diagnosis

The ASD diagnosis tends to occur after the preschool years [31,32,33]. Thus, establishing a definitive diagnosis before the age of two to three is difficult because of the changeability in infant development and the wide variety of disorders that can be associated with the same early impairments [10,14,34,35,36]. Learning more about how and when these early deficiencies begin to appear can significantly contribute to elucidating when the first manifestations of ASD take place, increasing the possibility of carrying out an early intervention [9,37].

The main objective of the present longitudinal study was to evaluate early joint attention in a sample of 32 children born in the province of Valencia (Spain) at two developmental moments (8 and 12 months) in order to explore whether the alterations in this early communication skill are related to and predict the presence of risk markers for ASD at 18 months of age. This was a study that replicated some of the contributions that have been made in recent years on joint attention and early detection of autism in a Spanish population. For the most part, the contributions that have been made in this regard have taken place in the Anglo-Saxon population. In this case, we wanted to test the presence of early risk indicators for ASD in the Spanish-speaking population in Spain to contribute to the issue in another socio-linguistic context.

## 2. Materials and Methods

### 2.1. Participants

To recruit the sample, several professionals from different health centers and Hospitals in the city of Valencia (Spain) collaborated. The different professionals informed the families who attended their service about the project and offered them the opportunity to participate. Those families that were willing to participate in the research provided their contact information so that they could be telephoned by the research team. The children participating in this study were those whose families accepted the pediatrician’s invitation to participate in it. They were healthy children without any special medical conditions. The eligibility criteria to be informed by the pediatrician were: (1) absence of any medical condition associated with an increased risk of developing a neurodevelopmental disorder; (2) absence of a neurodevelopmental or metabolic developmental disorder; (3) absence of pre, peri, and/or postnatal complications. Those children whose families agreed to participate in the study were the ones who constituted the sample. Therefore, this was a causal or incidental sample.

The final sample consisted of 32 babies assessed at 8, 12, and 18 months of age. Of these children, 12 were boys and 20 were girls. The average age of the mothers of the participating babies was 35.71 years, and the average age of the fathers was 37.19 years. More characteristics about the participating babies and their families can be seen in Table 1.

The families’ participation in this study was voluntary, anonymous, and non-remunerated. All the families were duly informed and in all cases, informed consent for the use of personal data and video images was signed.

### 2.2. Instruments and Measures

#### 2.2.1. Joint Attention

The instrument used to assess joint attention skills was the early social-communication scales (ESCS) [8]. This instrument is used to assess individual differences in pre-verbal communication skills that usually appear in children between 8 and 30 months of age. It has a duration of 15–20 min, and the whole session was taped to facilitate the identification of the target behaviors. Children’s behavior was coded in three categories: ‘joint attention’, ‘behavioral request’, and ‘social interaction’. Within these categories, it was also taken into account whether they were behaviors initiated by the child or if the child was responding to the examiner (see Table 2). Early social skills were evaluated through tasks in which the examiner presented different objects (toys) to the child. The measure obtained with this instrument was the frequency of appearance of each behavior.

For this research, only the dimensions of joint attention and behavioral request were used for the analyses. With the ESCS [8], the variables of initiating joint attention (IJA), responding to joint attention (RJA), initiating behavioral request (IBR), and responding to behavioral request (RBR) were measured. We must keep in mind that joint attention variables referred to ‘joint attention behaviors’ with declarative purposes, whereas behavioral request behaviors referred to ‘joint attention behaviors’ with imperative purposes.

To ensure the quality of the data obtained with the ESCS [8], encodings were carried out by two independent observers in 54.33% of the total registry. The interrater agreement was calculated for each dimension of the instrument. The average reliability of the data (intraclass correlation) was 0.768 for 8 month measures and 0.814 for 12 month measures.

#### 2.2.2. Early ASD Symptomatology

The risk of ASD at 18 months, indicated by the presence or absence of markers, was obtained by administering the M-CHAT (modified checklist for autism in toddlers) [38]. The M-CHAT [38] is a 23-item parent questionnaire with yes and no answer options for the identification of early behaviors associated with ASD screening for children between 16 and 30 months of age. This is a free tool for clinical, research, and training purposes.

Some of the questions included in the M-CHAT [38] are: “Does your child ever use his or her index finger to point or ask for something?”; “Does he or she bring objects to show them to you?”; “Does your child ever seem oversensitive to noise? (for example, reacts by covering his/her ears, etc.)”; “Is your son or daughter interested in other children? (for example, looks at other children, smiles at them, or approaches them)”; “Does your child respond when called by name? (for example, turns his/her head, talks or stutters, or stops what he/she was doing to look at you)”.

Based on the number of risk-rated items, levels can be established: *low risk* (between 0 and 2 points); *medium risk* (between 3 and 7 points); *high risk* (between 8 and 20 points). In the present research, we made this a dichotomous variable. Thus, a cut-off point was set at 2 points: “low risk” and “risk”. The variable RISK18, which refers to early ASD markers, was established from the score on the M-CHAT [38].

### 2.3. Procedure

The different evaluation sessions in this study were conducted in the research laboratories of the Faculty of Psychology of the University of Valencia (Spain). This was done to ensure that the evaluations were carried out in a controlled situation that was the same for all the participants. Each assessment session lasted approximately one hour. During the session, only the baby, the mother or father, and an examiner were present in the assessment room. The baby was accompanied by a caregiver at all times.

All the sessions were videotaped because some of the instruments used were for observational recording and required visualization of the images in order to accurately identify target behaviors. The data, both image and sound, were kept by the team in accordance with current Spanish regulations on the protection of personal data (LO 15/1999 and LO 3/2018). All of this was fully reported to the families and stated in the informed consent.

### 2.4. Ethics

This research was the result of a collaboration with the University and Polytechnic Hospital “La Fe” in the city of Valencia (Spain). The project was considered by the hospital’s Ethics Committee on Research on 9 May 2017 (2017/0167), which found that the project met the ethical requirements in terms of the design and dealing with participating families. At no time did this study include any type of invasive procedure for the health or physical or moral integrity of the minors or their companions.

### 2.5. Analysis

The ASD risk variable (RISK18) was a dichotomous variable in this study. For this reason, in order to explore the relationship between early joint attention and ASD risk markers, two multiple logistic regression analyses (backward stepwise method) were performed: one for the joint attention variables at 8 months of age and another for the same variables at 12 months. It should be mentioned that the behavioral request variable (RBR) was not included in the 8 month analysis because this variable referred to behaviors that are not yet observable at this young age. Thus, this variable showed no values at 8 months because the behaviors that the variable referred to are not observable until around 12 months of age. Analyses were performed on participants whose results for the variables used were obtained at all three times (8, 12, and 18 months). The total number of participants who were included in these analyses was 32. Of these 32 children, four obtained results indicating risk of ASD, according to the M-CHAT [38].

## 3. Results

### 3.1. Joint Attention at 8 Months and ASD Risk Markers at 18 Months of Age

The results obtained for this multiple logistic regression analysis are presented in Table 3. As we can see in last step, the IJA8 variable showed a result close to statistical significance in predicting ASD risk at 18 months (*p* = 0.051), and furthermore, it explained the 53.1% of the variance, according to Nagelkerke’s criterion.

### 3.2. Joint Attention at 12 Months and ASD Risk Markers at 18 Months of Age

The results obtained for this multiple logistic regression analysis are presented in Table 4. As we can see in last step, the RJA12 variable showed a statistically significant result in predicting ASD risk at 18 months (*p* = 0.014), and furthermore, it explained the 45.2% of the variance, according to Nagelkerke’s criterion.

## 4. Discussion and Conclusions

The present study was designed to examine the role of joint attention in predicting the risk of early ASD symptomatology in the first year of the baby’s life. Our results revealed that joint attention alterations at the ages of 8 and 12 months were associated with early ASD symptoms at 18 months old. IJA impairments at 8 months showed a prediction close to statistical significance for ASD risk at 18 months. Moreover, RJA impairments at 12 months significantly predicted early ASD symptomatology. These findings shed light on the approach to early detection and observation of first symptoms of ASD in a Spanish population. Our results are in line with those obtained with samples from other socio-linguistic contexts. In this sense, it can be said that the results were successfully replicated in a new sociolinguistic context.

These results agree with the findings of Sullivan et al. [39], who suggested that RJA impairments can be reliable indicators of early ASD symptomatology at 14 months old in an infant population of at-risk children. Our results also coincide with previous research highlighting the importance of IJA measurements in identifying early ASD symptomatology [13,21,30,40,41,42,43,44]. In addition, these results appear to support the idea that early ASD signs can be observed at the age of 8–9 months in some cases [9,13,41,42,45,46].

Nevertheless, it should be noted that some authors claim that not only are the measures of joint attention informative but so is the evolution of IJA. Although early measures of IJA are important in considering the existence of early signs of ASD, it is more important to observe the development of IJA. Its evolution is what actually helps to identify children who are more likely to receive the diagnosis [41,42]. Furthermore, authors such as Gangi et al. [47] and Landa et al. [12] recommend proceeding with caution when considering IJA measures as a definite indicator of early signs of ASD, especially at very young ages. Some children later diagnosed with ASD did not show any signs at 10–12 months old [48]. Some authors suggest that most at-risk infants show early ASD symptomatology at around 12–14 months [5,9,10], but these signs only become completely clear after the age of 18 months [19,20,21,22]. A design with more assessment times can explore this aspect in greater depth.

However, it can be said that our results are promising because early joint attention abilities at the ages of 8 months and 12 months did show to be linked to ASD early symptomatology at 18 months old. IJA and RJA variables were found to be almost significant and significant, respectively, in predicting early ASD signs at 18 months.

Both IJA and RJA have been associated with ASD early symptomatology in past research. Specifically, studies have shown that IJA disturbances are a better predictor of early ASD symptomatology than RJA deficiencies. In other words, a weaker presence of initiating behaviors constitutes a more reliable risk sign than a lower occurrence of responding behaviors [40,49,50,51,52]. It appears that early RJA deficiencies may abate only in older children with ASD when language has emerged. IJA impairments tend to be present during the preschool period and throughout adolescence [9,49,53]. Unlike these studies, we did find a link between IJA and RJA with early signs of ASD. IJA deficiencies at 8 months were predictors of ASD early markers at 18 months, with values close to significance; whereas RJA deficits at 12 months were predictors of ASD early signs at 18 months, with clearly significant values.

Finally, our results are consistent with the proposal that declarative non-verbal communication (*joint attention*, as opposed to *behavioral request*) is frequently more affected in children with autism than imperative non-verbal communication (*behavioral request*) [54,55,56]. Joint attention variables were significantly stronger predictors of the first signs of ASD, while the behavioral request variables showed little or no ability to predict the first ASD markers. Joint attention variables might be expected to significantly predict ASD early markers since impairments in joint attention are a characteristic of individuals with autism [21,43,44].

In conclusion, difficulties with joint attention at an early age are good indicators of early symptoms of ASD at 18 months, and numerous studies support this idea [21,43,44]. Difficulties in initiating joint attention at 8 months old can predict the existence of ASD early markers at 18 months, as other studies have also shown [41], and difficulties in appropriately responding to an invitation for joint attention at 12 months of age provide evidence of early symptomatology [45,57].

## 5. Limitations and Prospective

This research aimed to take an initial step in exploring the role of infants’ social communication in the early detection of ASD. We hope that these findings can contribute to identifying the first signs of ASD at very young ages, but we would like to acknowledge some aspects that should be taken into account when interpreting the results.

Firstly, this study had a relatively small sample, which constitutes a handicap since we understand that a small sample limits the scope of the results obtained. In addition, this study used an incidental sample. This means that only those children whose families agreed to participate in the study were the ones who constituted the sample. Therefore, we are calling for caution when considering the results since these group of babies might not accurately represent the population.

However, the difficulty of obtaining samples with these characteristics (very young children) and the high cost of longitudinal research designs should be kept in mind and valued. Longitudinal studies require a considerable time investment, and the initial results can only be observed in the medium or long term. In addition, it is necessary to consider the sample cost because maintaining the fidelity of the participants is a great challenge. For this reason, we are calling for future studies to attempt to replicate these results in a larger sample.

Another reason for interpreting the results with caution is that differences in social communication skills in early infancy may not always be detectable through general behavioral observation. Thus, measuring the frequency of the occurrence of a behavior may not always allow us to capture these phenomena. The set of early ASD signs can be further studied by incorporating potential biomarkers obtained from neuroelectrophysiology [58] and genetic research [59]. Numerous authors have highlighted the need to broaden the understanding of valid biometrics [60,61,62]. Research that includes more detailed techniques such as eye-tracking measures can be extremely informative [10,63,64]. Moreover, these methods can be used with infants as young as 6 months, allowing even earlier detection.

Finally, we must acknowledge the fact that this study did not explore aspects such as family socioeconomic status or education, which have shown to be of great importance when considering infant communicative skills. Literature widely reports that family income and parental education are two variables that reliably and significantly predict parent–child interaction and in turn, young children’s early communication and language skills [65,66]. Low-income parent–child dyads are associated with dysregulated interactions during the first year, which predicts young children’s communication skills after the first birthday. In addition, parental education usually mediates this relationship, strengthening it in the case of parents with little training and educational opportunities whose abilities to engage in sensitive and enriching interactions are lesser. In that sense, not having explored these aspects constitutes a limitation.

Despite the study’s limitations, we believe our findings are valuable and provide a better understanding of early social communication skills and joint attention as reliable predictors of ASD risk at very young ages. Even without a confirmation of the diagnosis, early intervention is especially important because it allows at-risk children to improve their social skills and consequently, their quality of life [10,22,67,68,69,70,71,72].

## Figures and Tables

**Figure 1 children-09-00556-f001:**
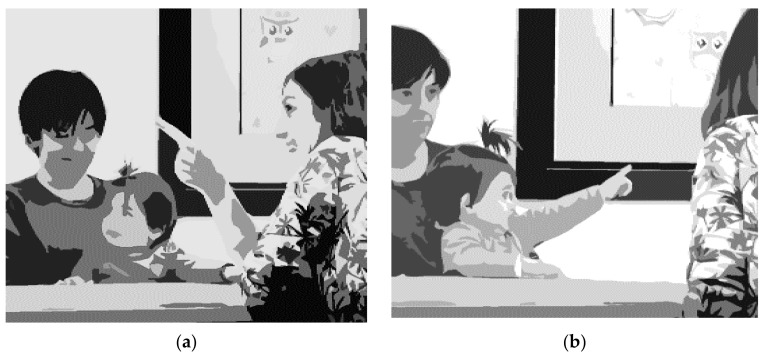
This figure illustrates some joint attention behaviors: (**a**) image of a child responding with a head-turn to a cue from the examiner (RJA); (**b**) image of a child pointing to a picture on the wall (IJA).

**Table 1 children-09-00556-t001:** Characteristics of the participants and their families.

	N	%		N	%
**Number of siblings**			**Family structure**		
Single child	15	46.9	Single parent	1	3.1
One sibling	15	46.9	Nuclear family	30	93.8
Two siblings	1	3.1	Reconstituted family	1	3.1
Three siblings	1	3.1	Extended family	0	0
**Standard of education of father**			**Standard of education of mother**		
No studies	1	3.1	No studies	0	0
Primary education	2	6.3	Primary education	2	6.3
Secondary education	7	21.9	Secondary education	5	15.6
Higher education	21	65.6	Higher education	25	78.1
**Employability of education of father**			**Employability of education of mother**		
Unemployed	2	6.3	Unemployed	9	28.1
Homemaker	0	0	Homemaker	0	0
Part-time job	3	9.4	Part-time job	6	18.8
Full-time job	25	78.1	Full-time job	15	46.9
Pensioner/Retiree	0	0	Pensioner/Retiree	1	3.1
Others	1	3.1	Others	1	3.1
**Context**			**Family income**		
Urban	22	68.8	EUR 12,000–23,999	8	25
Residential area	6	18.8	EUR 24,000–35,999	11	34.4
Rural	4	12.5	EUR 36,000–50,000	12	37.5
Others	0	0	EUR >50,000	1	3.1

**Table 2 children-09-00556-t002:** Early communication skills assessed with the ESCS.

Joint Attention	Behavioral Request	Social Interaction
Initiating joint attention (IJA)	Initiating behavioral request (IBR)	Initiating social interaction (ISI)
The child voluntarily orients the adult’s gaze towards an object or event by looking at or pointing to it.	The child orients the adult’s attention to objects because he/she wants to have them (or wants to get something from the adult).	The child initiates the episode of social interaction (for example, by grasping a ball and throwing it to the adult, with the expectation that the adult will throw it back).
Responding to joint attention (RJA)	Responding behavioral request (RBR)	Responding social interaction (RSI)
The child follows the invitation to participate in a joint attention episode by orienting his/her gaze toward the object or the event to which the adult is referring.	The child responds to simple gestural or verbal commands made by the adult to obtain an object or perform an action.	The child responds to the adult’s invitation to engage in a face-to-face social game.

**Table 3 children-09-00556-t003:** Results for multiple logistic regression analysis (8 months).

		*B*	Standard Error	Wald	*p*	Exp (*B*)	C.I. 95% for Exp (*B*)	
							Lower Limit	Upper Limit
Step 1	IJA8	−0.432	0.256	2.856	0.091	0.649	0.393	1.072
	RJA8	−0.012	0.034	0.112	0.738	0.989	0.924	1.058
	IBR8	−0.106	0.288	0.134	0.714	0.900	0.511	1.583
	Constant	2.584	0.192	1.390	0.238	13.244		
Step 2	IJA8	−0.436	0.253	2.958	0.085	0.647	0.394	1.063
	IBR8	−0.130	0.284	0.211	0.646	0.878	0.503	1.531
	Constant	2.290	1.998	1.313	0.252	9.870		
Step 3	IJA8	−0.471	0.241	3.822	0.051	0.625	0.390	1.001
	Constant	1.728	1.455	1.409	0.235	5.628		
Dependent variable: RISK18								

**Table 4 children-09-00556-t004:** Results for multiple logistic regression analysis (12 months).

		*B*	Standard Error	Wald	*p*	Exp (*B*)	C.I. 95% for Exp (*B*)	
							Lower Limit	Upper Limit
Step 1	IJA12	0.153	0.124	1.504	0.220	1.165	0.913	1.487
	RJA12	−0.099	0.063	2.473	0.116	0.905	0.800	1.025
	IBR12	−0.189	0.208	0.826	0.363	0.828	0.551	1.244
	RBR12	−0.074	0.077	0.924	0.336	0.928	0.797	1.080
	Constant	4.478	4.395	1.038	0.308	88.031		
Step 2	IJA12	0.130	0.121	1.150	0.284	0.138	0.898	1.443
	RJA12	−0.099	0.052	3.556	0.059	0.906	0.817	1.004
	RBR12	−0.086	0.066	1.692	0.193	0.917	0.805	1.045
	Constant	2.934	2.795	1.102	0.294	18.810		
Step 3	RJA12	−0.075	0.038	3.962	0.047	0.928	0.862	0.999
	RBR12	−0.056	0.048	1.346	0.246	0.946	0.861	1.039
	Constant	4.010	2.506	2.562	0.109	55.153		
Step 4	RJA12	−0.076	0.031	6.020	0.014	0.927	0.872	0.985
	Constant	2.776	1.763	2.480	0.115	16.052		
Dependent variable: RISK18								

## Data Availability

The data is available from the corresponding author.

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
