# Peer review of "Joint Attention and Its Relationship with Autism Risk Markers at 18 Months of Age"

_children, 2022, doi:10.3390/children9040556_

Round 1

Reviewer 1 Report

Major comments

While the study report and the presented statistical analyses are fine, the chosen statistical method  of logistic regression is not used exhaustively. The presented analyses  are only univariate throughout and thus do not show the additive predictive power of linear combinations of simultaneously available regressors which can be evaluated by using multiple regression models.

To better use the available information, multiple regression models should be selected by backwards or forward search among the available 8 months and 12  months predictors, and possibly among 8 months and 12  months predictors simultaneosly as the 8 months results may also be available when 12 months measurements were determined.

A possible alternative to the use  of RISK18 (binary) as an endpoint for logistic regression analysis would be the use of M-CHAT (continuous) as an endpoint for ordinary least squares regression analysis. The latter may be more powerful. This OLS regression analysis should be considered as  main analysis model or sensitivity analysis.

Minor comments

It is not clear what the selection criterion for inclusion into the study was. Obviously, it was not a random sample of healthy children, but what exactly where the conditions for children to qualify for the study? Take this point into regard in the limitations section.

Table 3: Why was RBR8 not evaluated? Add the reason, please, or add the predictor, if available.

Reviewer 2 Report

please see attached.

The comments are mentioned in the text

Round 2

Reviewer 1 Report

The changes to the paper met my comments. Thank you.